# Improved Low-cost 3D Reconstruction Pipeline by Merging Data From Different Color and Depth Cameras

Category: Research

**ABSTRACT**

The performance of traditional 3D capture methods directly influences the quality of digitally reconstructed 3D models. To obtain complete and well-detailed low-cost three-dimensional models, this paper proposes a 3D reconstruction pipeline using point clouds from different sensors, combining captures of a low-cost depth sensor post-processed by Super-Resolution techniques with high-resolution RGB images from an external camera using Structure-from-Motion and Multi-View Stereo output data. The main contribution of this work includes the description of a complete pipeline that improves the stage of information acquisition and makes the data merging from different sensors. Several phases of the 3D reconstruction pipeline were also specialized to improve the model's visual quality. The experimental evaluation demonstrates that the developed method produces good and reliable results for low-cost 3D reconstruction of an object.

**Keywords:** Low-Cost 3D Reconstruction, Depth Sensor, Photogrammetry.

**Index Terms:** Computer graphics, Shape modeling, 3D Reconstruction.

## 1 INTRODUCTION

3D reconstruction makes it possible to capture the geometry and appearance of an object or scene allowing us to inspect details without risk of damaging, measure properties, and reproduce 3D models in different material [21]. In recent years, numerous advances in 3D digitization have been observed, mainly by applying pipelines for three-dimensional reconstruction using costly high-precision 3D scanners. In addition, recent researches have sought to reconstruct objects or scenes using depth images from low-cost acquisition devices (e.g., the Microsoft Kinect sensor [17]) or using Structure from Motion (SFM) [24] combined with Multi-View Stereo (MVS) [5] from RGB-images.

Good quality 3D reconstructions require a large number of financial resources, as they require state-of-the-art equipment to capture object data in high precision and detail. On the other hand, low-resolution equipment implies a lower quality capture, even being financially more viable. Even with the ease of operation, lightweight, and portability, low-cost approaches must consider the limitations of the scanning equipment used [20].

The acquisition step of a 3D reconstruction pipeline refers to the use of devices to capture data from objects in a scene such as their geometry and color [22]. One result of 3D geometry capture is the production of discrete points collection that demonstrates the model shape. We call it point clouds. The data obtained by this step will be used in all other phases of the 3D reconstruction process [2].

Active capture methods use equipment such as scanners to infer objects geometry through a beam of light, inside or outside the visible spectrum. The scanner sensor has the advantages of fast measuring speed, robustness regarding external factors, and ease of acquiring information. Active sensors also have good performance in reconstructing texture-less and featureless surfaces [6, 22]. The sensors need to be sensitive to small variations in the information acquired, since for small differences in distance, the variation in the time it takes to reach two different points is very low, requiring low equipment latency and good response time. For this reason, these

systems tend to be slightly noisy [21]. Considering low-cost reconstruction approaches difficulties to capture color in high precision are disadvantage [10].

Passive methods are based on optical imaging techniques. They are highly flexible and work well with any modern digital camera. Image-based 3D reconstruction is practical, non-intrusive, low-cost and easily deployable outdoors. Various properties of the images can be used to retrieve the target shape, such as material, viewpoints and illumination. As opposed to active techniques, image-based techniques provide an efficient and easy way to acquire the color of a target object [10]. Although passive reconstructions mainly using SFM and MVS produce excellent results, they have limitations like the difficulty of distinguishing the target object from the background [25] and require the target object to have detailed geometry [6]. A controlled environment is needed to obtain better reconstruction results [12, 24].

Considering the limitations imposed by the presented approaches, it is important to note that a target whose geometry has been described by only a low-cost capture method has a real challenge in expressing its completeness, with rich and small details [6].

This paper proposes a hybrid pipeline from a low-cost depth camera (low-resolution images) and an external color capture camera (digital camera with high-resolution RGB images) to estimate and reconstruct the surface of an object and apply a high-quality texture. Such limitations of each data acquisition approach are bypass, generating a complete and well-detailed replica of the target model with high visual quality. To achieve this effect, this project uses a variation and combination of Structure from Motion, Multi-View Stereo and depth camera capture techniques.

Although there are mature projects aimed at low-cost 3D reconstruction, few are those who describing step-by-step how to overcome the limitations from low-cost three-dimensional data capture using the best features in all phases of the pipeline to obtain the model as realistic as possible. The main contribution of this work is the description of a complete pipeline that makes use of post-processed depth captures and merging data from different sensors, in which depth sensor data and high-resolution color images do not need to be synchronized.

As it is a post-processed task (after capture/estimate depth data), this work also includes the detection of the region of interest, based on the average distance of the scene, removing points not belonging to the target object and allows the inclusion of new images containing regions of the target object not previously photographed to improve the texturing step results.

In addition to this introductory section, this work is organized as follows: Section 2 presents related works, while section 3 describes the proposed pipeline. The experiments and evaluation of the pipeline are presented in section 4. Finally, section 5 discusses the final considerations and results achieved by this research.

## 2 RELATED WORK

Prokos et al. [19] proposed a hybrid approach combining shape from stereo (with additional geometric constraints) and laser scanning techniques. Using two cameras and a portable laser beam, they achieved accuracy as good as some high-end laser triangulation scanners. Although, they do not include automatically detecting outliers in their results.

The KinectFusion system [17] tracks the pose of portable depth cameras (Kinect) as they move through space and perform good three-dimensional surface reconstructions in real-time. The Kinect sensor has considerable limitations, including temporal inconsistency and the low resolution of the captured color and depth images [22]. Real-time reconstruction is not a requirement for well-detailed, accurate, and complete reconstructions.

Silva et al. [26] provides a guided reconstruction process using Super-Resolution (SR) techniques, helping to increase the quality of the low-resolution data captured with a low-cost sensor. The method of data acquisition using low-cost depth cameras and SR is also improved by Raimundo [22]. Even with depth image improvements, a poor registration of captures can affect the final model's shape.

Falkingham [9] demonstrates the potential applications of low-cost technology in the field of paleontology. The Microsoft Kinect was used to digitalize specimens of various sizes, and the resulting digital models were compared with models produced using SFM and MVS. The work pointed out that although Kinect generally registers morphology at a lower resolution capturing less detail than photogrammetry techniques, it offers advantages in the speed of data acquisition and generation of the 3D mesh completed in real-time during data capture. Also, they did not use Super-Resolution to improve captures from low-cost devices and the models produced by the Kinect lack any color information.

Zollhöfer et al. [28] used a Kinect sensor to capture the geometry of an excavation site and took advantage of a topographic map to distort the reconstructed model, significantly increasing the quality of the scene. The global distortion, with Super-Resolution techniques applied to raw scans, significantly increased the fidelity and realism of its results but is too specialized for large scales scenes.

Paola and Inzerillo [8] in order to digitally produce the Egyptian stone from Palermo, proposed a method with a structured light scanner, smartphones and SFM to apply texture in the highly accurate mesh generated by the scanner. The main challenges were the dark color of the material and the superficiality of the groove of the hieroglyphs that some capture approaches have difficulty recognizing. The level of detail of the texture application showed up quite accurately. This reference work used a high-resolution 3D scanner, not aiming at low-cost reconstruction.

Jo and Hong [13] use a combination of terrestrial laser scanning and Unmanned Aerial Vehicle (UAV) photogrammetry to establish a three-dimensional model of the Magoksa Temple in Korea. The scans were used to acquire the perpendicular geometry of buildings and locations, being aligned and merged with the photogrammetry output, producing a hybrid point cloud. The photogrammetry adds value to the 3D model, complementing the point cloud with the upper parts of buildings, which are difficult to acquire through laser scanning.

Chen [6] proposes a registration method to combine the data of a laser scanner and photogrammetry to reconstruct the real outdoor 3D scene. They managed greatly increasing the accuracy and convenience of operation. The two sensors can work independently, the method fuses their data even if in different scales. Mesh reconstruction and texturing were not explored by this work.

Raimundo et al. [21] point out in their bibliographic review several studies that successfully used advanced rendering techniques such as global illumination, ambient occlusion, normal mapping, shadow baking, per-vertex lighting, and level of detail. These rendering techniques also improve the final presentation of 3D reconstructions.

## 3 PIPELINE PROPOSAL

To overcome limitations of the low-cost three-dimensional data acquisition process, the following pipeline is proposed: capturing depth and color images (using a low-cost depth sensor and a digital camera); generation of point clouds from low-cost RGB-D camera

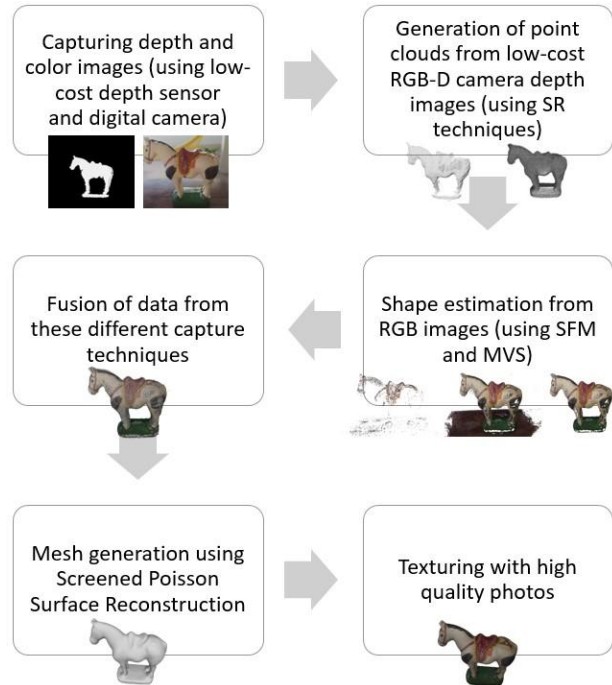

Figure 1: Schematic diagram for the proposed pipeline and the 3D reconstruction processes of an object.

depth images (using SR techniques [22]); shape estimation from RGB images (using SFM [24] and MVS [5]); merging of data from these different capture techniques; mesh generation; and texturing with high quality photos (Fig. 1). Several phases of the pipeline were specialized to achieve better accuracy and visual quality of 3D reconstructions of small and medium scale objects. The proposed pipeline works offline.

### 3.1 Data acquisition

For capture using a low-cost depth sensor is established the following acquisition protocol: take several depths captures, moving the sensor around the object, and defining the limits of the capture volume. Furthermore, a turntable can also be used, obtaining a more controlled capture and align process. The number of views captured is less than that of real-time approaches due to the additional processing required to ensure the quality of each capture. Considering the quality requirements for this proposed work, an interactive tool [20] is used to acquire the raw data from the depth sensor (Fig. 2).

The depth capture method will present results proportional to the better the captures by the device, that is, the lower the incidence of noise and the better the accuracy of the inferred depth. With this in mind, each depth image goes through a filtering step with the application of Super-Resolution [22]. To provide high-resolution information beyond what is possible with a specific sensor, several low-resolution captures are merging, recreating as much detail as possible.

To add 3D information in greater detail and to apply a simple high-quality texturing process, photographs are taken from a digital camera around the target object. In our pipeline, these captures are independent of the depth sensor, we need just to take pictures with the fixed object, in a free movement of the camera. The set of images must be sufficient to cover most of the object's surface and the images must portray, in pairs, common parts of it. The color images will be used in the SFM pipeline.

The SFM pipeline detects characteristics in the images (feature de-

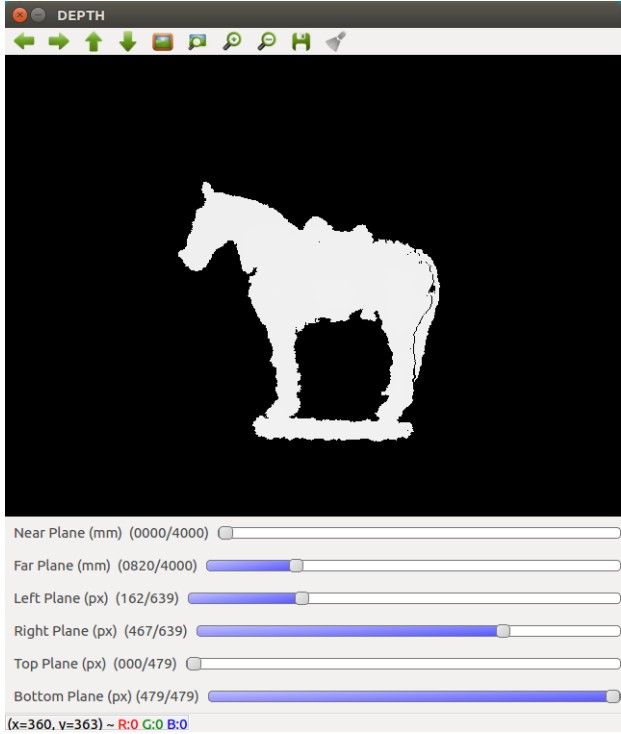

Figure 2: Software to acquire and process depth images. The slider controls the capture limits (in millimeters) and the cut limits (in pixels), effectively determining the capture volume.

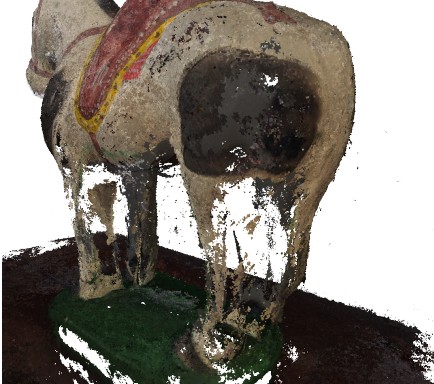

(a) MVS point cloud result

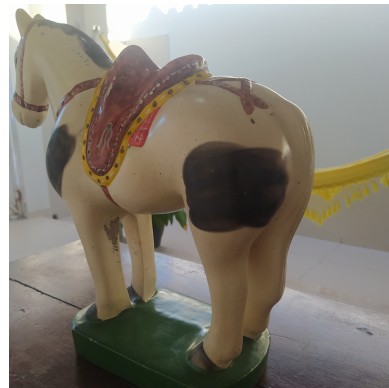

(b) RGB photo of porcelain horse

Figure 3: Some parts of the surface may not be estimated by the photogrammetry process. In (a) the white and smooth painting of the object (b) prevents the MVS algorithm from obtaining a greater number of points that define this part of the structure of the model, leaving this featureless surface region with a fewer density of points than others.

tection), mapping these characteristics between images and finding descriptors capable of representing a distinguishable region (feature matching). These descriptors represent vertices of the reconstruction of the 3D scene (sparse reconstruction). The greater the number of matches found between the images, the greater the degree of accuracy of calculating a 3D transformation matrix between the images, providing the estimation of the relative position between camera poses [3, 10].

Photographs with good resolution and objects with a higher level of detail tend to bring greater precision to the photogrammetry algorithms. For objects with fewer details and features, the environment can be used to achieve better results [24]. In addition to the estimated structure to improve the depth sensor captured geometry, we use these cameras' pose estimation to apply easily and directly texture over the final model surface.

The Multi-View Stereo process is used to improve the point cloud obtained by SFM, resulting in a dense reconstruction. As the camera parameters such as position, rotation, and focal length are already known from SFM, the MVS computes 3D vertices in regions not detected by the descriptors. Multi-View Stereo algorithms generally have good accuracy, even with few images [10].

For this image-based point cloud result, to highlight the target object, a method of detecting the region of interest can be used. A simple algorithm is used to detect the centroid of the set of 3D points and remove points based on a radius from it. If the floor below the object is discernible, it is also possible to use a planar segmentation algorithm to remove the plane. A statistical removal algorithm can also be used to remove outliers. If even more accurate outlier removal is required, a manual process using a user interface tool can be performed. Most of the discrepancies and the background are removed using the proposed steps, minimizing working time.

Although image-based 3D reconstructions get greater detail than using low-cost depth sensors [9], this approach may not be able to estimate the completeness of the object (Fig. 3). This is a common result when the captures do not fully describe the target model, or it does not have a very distinguishable texture or detail.

The algorithms used in the next steps require a guided set of data, thus, the normals of the point clouds are estimated before performing the alignment step. A normal estimation k-neighbor algorithm is used for this task.

## 3.2 Alignment

To deal with the problem of aligning the point clouds of the acquisition phase, transformations are applied to place all captures in a global coordinate system. This alignment is usually performed in a coarse and fine alignment step.

To perform the initial alignment between the point clouds obtained by the depth sensor we use global alignment algorithms where the pairs of three-dimensional captures are roughly aligned [15]. Given the initial alignment between the captured views, the Iterative Closest Point (ICP) algorithm [11] is executed to obtain a fine alignment. After pairwise incremental registration, an algorithm for global minimization of the accumulated error is run.

The initial alignment step may not produce good alignment results due to the nature of the depth data utilized, as the low amount of discernible points between two point clouds [20], so the registration may present drifts. With this in mind, we use the point cloud obtained by photogrammetry as an auxiliary to apply a new alignment

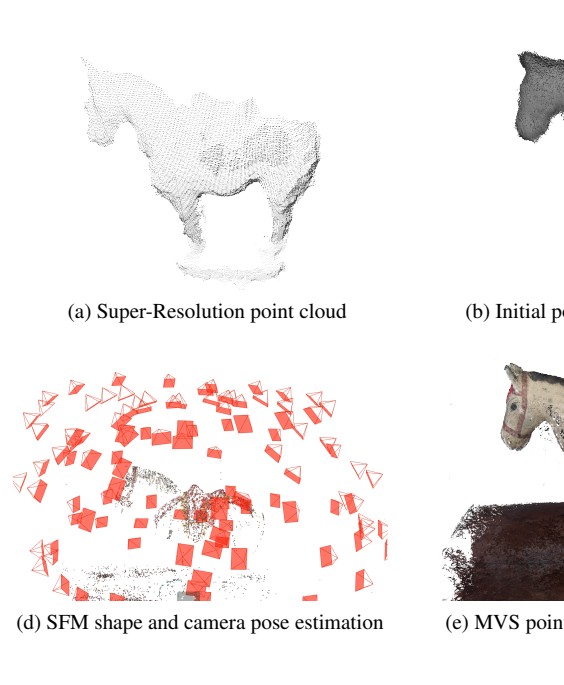

(a) Super-Resolution point cloud

(b) Initial point clouds alignment (Kinect)

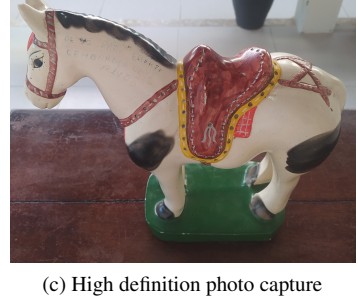

(c) High definition photo capture

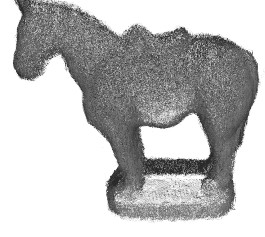

(d) SFM shape and camera pose estimation

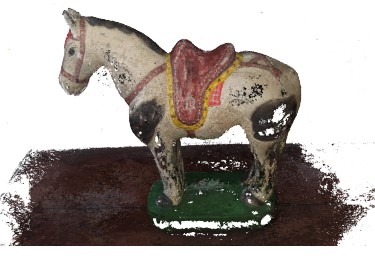

(e) MVS point cloud output before filtering

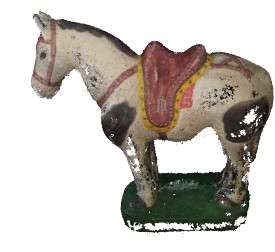

(f) MVS point cloud output after filtering and down-sampling

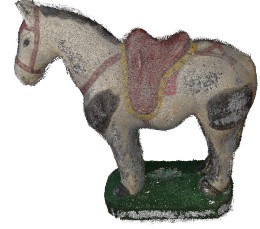

(g) Improved initial alignment and cleaning

(h) Merged point cloud

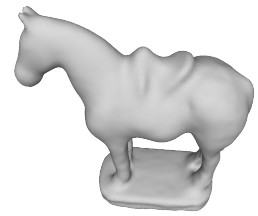

(i) Poisson generated mesh

Figure 4: Porcelain horse. With the richness of details that this object has, as in the head and saddle, we use the photogrammetry method for distinguishing them with the highest level of detail. At the same time it has a low number of characteristics in predominantly smoothness regions, as the base of the structure and the body of the animal, we use the depth sensor capture approach where this factor does not influence the 3D acquisition process. The data captured by the low-cost depth sensor aggregated information where there are few visible features, as can be seen at the base and legs of the horse.

over the depth sensors point clouds, distorting the transformation, propagating the accumulation of errors between consecutive alignments and the loop closure, improving the global registration and the quality of the aligned point cloud.

The point cloud generated by the image-based 3D reconstruction pipeline and the one obtained with the depth sensor captures are created from different image spectrum and are very common to have different scales. The point clouds obtained using the depth sensor must be aligned with the corresponding points of the object in the photogrammetry point cloud.

As the depth sensor captures are already in a global coordinate system, to carry out this alignment, it is sufficient just to scale and transform a single capture to fit the cloud obtained by MVS and apply the same transformation to the others, speeding up the registration process. After that, the ICP algorithm can be reapplied, including the photogrammetry output point cloud. This last point cloud is not to be transformed, only the rest of the captures is aligned to it because the camera positions that we will utilize for texturing will use this model's coordinate system.

The merging of point clouds from both data capture approaches will increase the information that defines the object geometry. This resulting point cloud is used on the next steps of the pipeline.

### 3.3 Surface reconstruction

The mesh generation step is characterized by the reconstruction of the surface, a process in which a 3D continuous surface is inferred from a collection of discrete points that prove its shape [1].

For this step, we use the algorithm Screened Poisson Surface Reconstruction [14]. This algorithm seeks to find a surface in which the gradient of its points is the closest to the normals of the vertices of the input point cloud. The choice of a parametric method for the surface reconstruction is justified by the robustness and the possibility of using numerical methods to improve the results. Also, the resulting meshes are almost regular and smooth.

### 3.4 Texture synthesis

Applying textures to reconstructed 3D models is one of the keys to realism [27]. High-quality texture mapping aims to avoid seams, smoothing the transition of an image used for applying texture and its adjacent one [16].

The texture synthesis phase of the proposed pipeline comprises the combination of the high-resolution pictures captured with an

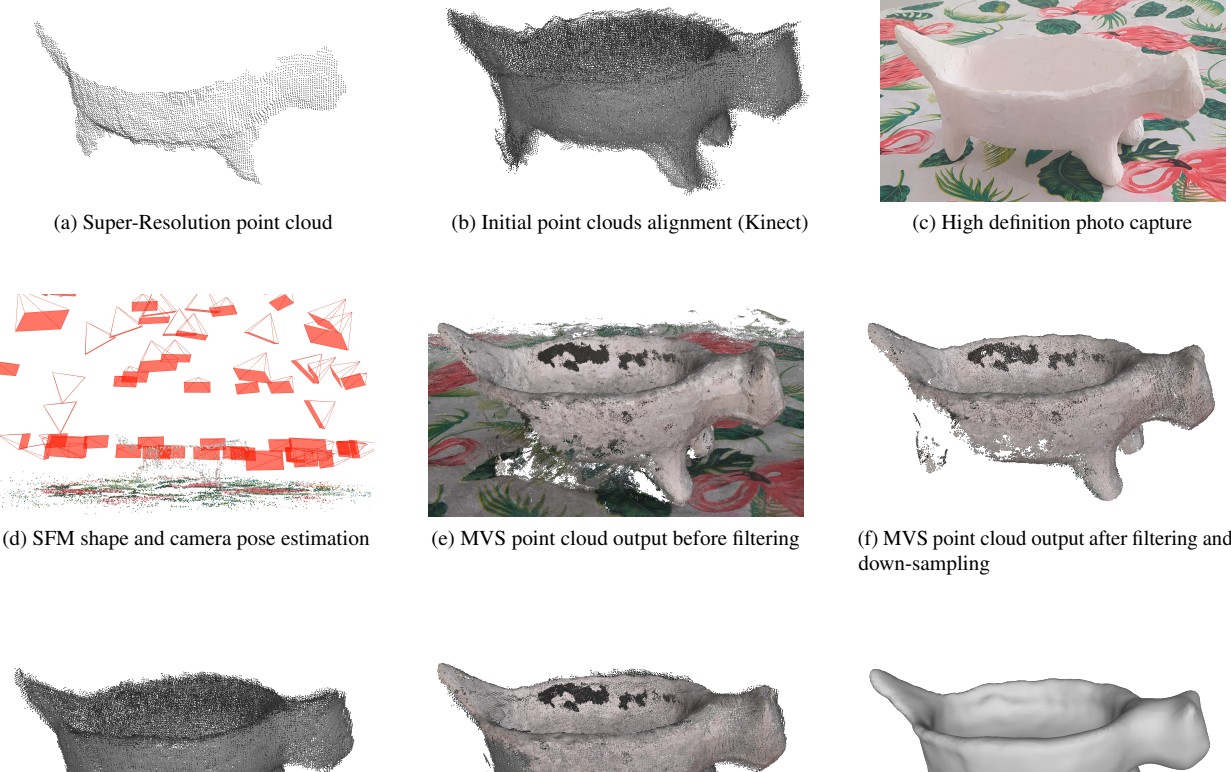

(a) Super-Resolution point cloud     (b) Initial point clouds alignment (Kinect)     (c) High definition photo capture

(d) SFM shape and camera pose estimation     (e) MVS point cloud output before filtering     (f) MVS point cloud output after filtering and down-sampling

(g) Improved initial alignment and cleaning     (h) Merged point cloud     (i) Poisson generated mesh

Figure 5: Jaguar pan replica. Even with some visual characteristics generated by the 3D printing process, the object has very few distinguishable features because of its predominantly white texture. This factor makes difficult the reconstruction process by SFM and MVS. With this, we use the environment to assist in detecting the positions and orientation of the cameras. The captures with the depth sensor added information in the legs of the jaguar and the belly (bottom) not acquired by photogrammetry.

external digital camera with the integrated model obtained from the previous step of the pipeline.

The high-resolution photos taken with a digital camera with the poses calculated using SFM, will be used to perform the generation of texture coordinates and atlas of the model, avoiding a time-consuming manual process.

The images with respective poses from SFM may not be able to apply a texture on faces not visible by any image used for the reconstruction, causing non-textured mesh surfaces in the three-dimensional model. To overcome this limitation, we post-apply the texture, merging camera relative poses result from SFM with new photos, calculating the new poses using photogrammetry result relative coordinate system.

## 4 EXPERIMENTS AND EVALUATION

For evaluation, we run the proposed pipeline on some objects varying size and complexity: a porcelain horse-shaped object ("Porcelain horse", Fig. 4), a jaguar and a turtle-shaped clay pan replicas ("Jaguar pan", Fig. 5 and "Turtle pan", Fig. 6 respectively). The remnant objects used in this study are replicas of cultural objects from the Waurá tribe and belong to the collection of Federal University of Bahia Brazilian Museum of Archaeology and Ethnology (MAE/UFBA). The replicas were three-dimensionally reconstructed

by Raimundo [20] and 3D printed. In addition, the turtle replica was colored by hydrographic printing.

In our experiments we used Microsoft Kinect version 1, however, any other low-cost sensor can be used to capture depth images. This sensor is affordable and captures color and depth information with a resolution of 640x480 pixels. To produce point clouds from the low-cost 3D scanner, we used the Super-Resolution approach proposed by Raimundo [22] with 16 Low-Resolution (LR) depth frames.

The photos used as input to the passive 3D reconstruction method were taken with a Redmi Note 8 camera for all evaluated models. The number of photos was arbitrarily chosen to maximize coverage of the object. For the SFM pipeline, the RGB images were processed using COLMAP [24] to calculate camera poses and sparse shape reconstruction. OpenMVS [5] was used for dense reconstruction. For the texturing stage, we used the algorithm proposed by Waechter et al. [27].

Some software tools were developed from third-party libraries for various purposes. For instance, OpenCV [4] and PCL [23] were used to handle and process depth images and point clouds, libfreenect [18] was used on the depth acquisition application to access and retrieve data from the Microsoft Kinect. Meshlab system [7] has been used for Poisson reconstruction and adjustments in 3D point clouds and meshes when necessary.

Table 1: Algorithms and main components of each experiment.

| Object | Porcelain horse | Jaguar pan | Turtle pan |
|---|---|---|---|
| **Dimensions (cm)** | 35 x 12 x 31 | 21.5 x 15 x 7 | 9 x 6.5 x 3.5 |
| **Texture** | Handmade | Predominantly white | Hydrographic printing |
| **Num. of RGB images** | 108 | 65 | 29 |
| **RGB images resolution** | 8000 x 6000px | 4000 x1844px | 8000 x 6000px |
| **SFM algorithm** | COLMAP [24] | COLMAP [24] | COLMAP [24] |
| **MVS algorithm** | OpenMVS [5] | OpenMVS [5] | OpenMVS [5] |
| **Depth sensor** | Kinect V1 | Kinect V1 | Kinect V1 |
| **LR frames per capture** | 16 | 16 | 16 |
| **SR point clouds** | 26 | 22 | 20 |

The Figures 4 and 5 show the acquisition, merging, and reconstruction steps proposed by this pipeline for the Porcelain Horse and Jaguar Pan. The figures also bring the discussion of the main challenges for each reconstruction and how they were handled by the pipeline. The algorithms and main components of each experiment are described in Table 1.

The resolution of clouds obtained by the low-cost sensor with SR is considerably lower than in clouds obtained by photogrammetry. This is evident in the turtle's captures and reconstructions (Fig. 6(b)). In such figure, is shown that the low-cost sensor presented a scale limitation. However, it has the advantage of making new captures of the object even if it has moved in the scene. The photogrammetry also presented limitations when it try to describe featureless regions of any object (as shown in Fig. 3 and Fig. 5(f)). However, this does not happen with the depth sensor since the coloring does not influence on capture. The resolution of the images used on the SFM pipeline is also a factor that directly influences the quality and details of the 3D reconstruction. The point clouds obtained by photogrammetry were capable of representing, with good quality, distinguishable details on a millimeter scale. The merging of point clouds was helpful to express in greater detail the objects that were reconstructed, taking the advantages of both captures.

The merged point clouds have been down-sampled to facilitate visualization and meshing generation since the aligned and combined point clouds may have an excessive and redundant number of vertices and there is no guarantee that the sampling density is sufficient for proper reconstruction [2]. Point clouds were meshed using the Screened Poisson Surface Reconstruction feature in Meshlab [7] using reconstruction depth 7 and 3 as the minimum number of samples. It is important to note that the production of a mesh is a highly dependent process on the variables used to generate the surface. We will consider as standard for all reconstructions the Poisson Surface Reconstruction the parameters defined in this paragraph.

For quantitative validation, the 3D surfaces reconstructions of the Turtle (Fig. 6) were compared with the model used for 3D printing (ground truth in Fig. 6(d)). For this comparison, we used the Hausdorff Distance tool of Meshlab [7]. The results are discussed on Table 2 and graphically represented on Fig. 7.

The same quantitative validation was carried out with the reconstructions of the Jaguar's 3D surfaces and its respective model used for 3D printing. The results are presented in Table 3 and as like the turtle's Hausdorff Distances, the reconstruction of the jaguar with this pipeline achieves better mean and lower values of maximum and minimum when compared with individual approaches.

All objects studied benefited from the merging of point clouds as Poisson's surface reconstruction identifies and differentiates nearby geometric details, some of them are added by the merging. It was noticed that, when the points are linearly spaced, the resulting mesh is smoother and more accurate.

Table 2: Hausdorff Distances for 3D surface reconstructions of the Turtle pan. Each vertex sampled from the source mesh is searched to the closest vertex on ground truth. Values in the mesh units and concerning the diagonal of the bounding box of the mesh.

| Mesh | MVS (Filtered) | Kinect (SR) | Merged |
|---|---|---|---|
| **Samples** | 17928 pts | 20639 pts | 20455 pts |
| **Minimum** | 0.000000 | 0.000003 | 0.000000 |
| **Maximum** | 0.687741 | 0.172765 | 0.124484 |
| **Mean** | 0.026021 | 0.028209 | 0.012780 |
| **RMS** | 0.082436 | 0.038791 | 0.023629 |
| **Reference** | Fig. 6(a) | Fig. 6(b) | Fig. 6(c) |

Table 3: Hausdorff Distances for 3D surface reconstructions of the Jaguar pan.

| Mesh | MVS (Filtered) | Kinect (SR) | Merged |
|---|---|---|---|
| **Samples** | 12513 pts | 13034 pts | 13147 pts |
| **Minimum** | 0.000005 | 0.000002 | 0.000001 |
| **Maximum** | 0.750001 | 0.173569 | 0.139575 |
| **Mean** | 0.051147 | 0.017597 | 0.019753 |
| **RMS** | 0.091608 | 0.028266 | 0.026867 |

Texturing results using surfaces from merged point clouds are shown in Fig. 8. This stage is satisfactory due to the high quality of the images used and from the camera positions correctly aligned and undistorted with the target object from SFM results.

The images with respective poses used by the SFM system did not be able to apply a texture on the bottom of the objects since bottom view was not visible. A new camera pose was manually added with the image of the bottom view on the SFM output, re-applying the texturing on this uncovered angle.

Every procedure described in this section was performed on a notebook Avell G1550 MUV, Intel Core i7-9750H CPU @ 2.60GHz x 12, 16GB of RAM, GeForce RTX 2070 graphics card, on Ubuntu 16.04 64-bits.

## 5 CONCLUSION

With the proposed pipeline, it is possible to add 3D capture information, reconstructing details beyond what a single low-cost capture method initially provides. A low-cost depth sensor allows preliminary verification of data during acquisition. The Super-Resolution

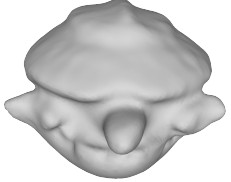 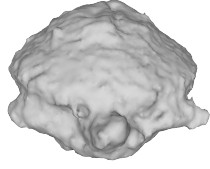 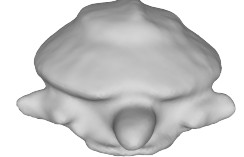 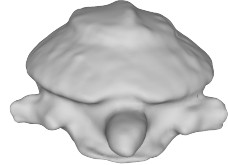

(a) Reconstruction from MVS output after filtering and down-sampling

(b) Reconstruction from SR Kinect captures with improved alignment

(c) Reconstruction from merged point clouds

(d) Ground truth mesh

Figure 6: Screened Poisson Surface Reconstruction results for the Turtle pan point clouds. The reconstruction depth is 7, while the minimum number of samples is 3 for all experiments. In (a) the limiting factor was the bottom part of the object that is not inferred by the photogrammetry process. (b) shows that the low-cost depth sensor was unable to identify details of the model, this is due to the small size of the object, making it difficult to obtain details, however, this mesh was able to represent the model in all directions, including the bottom. The merged mesh (c) was able to reproduce all the small details found by photogrammetry and include regions that were represented only by depth sensor captures. For comparison (d) presents the model's ground truth used for 3D printing.

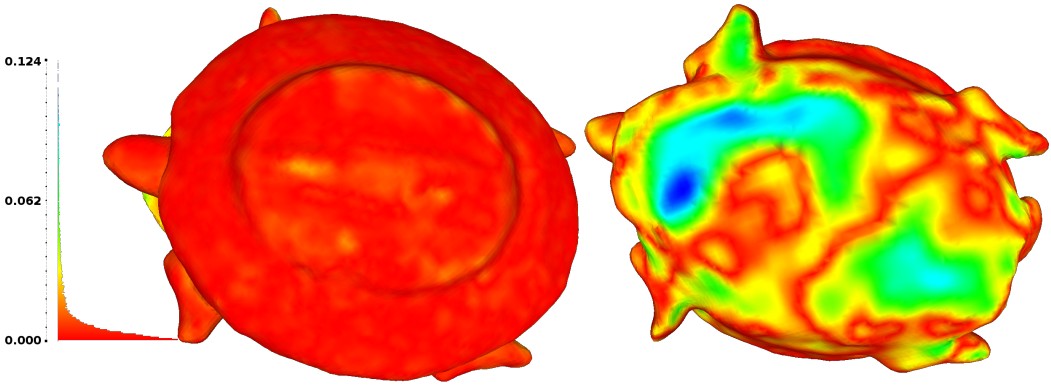

Figure 7: Hausdorff Distance of Turtle pan mesh result using the proposed pipeline (Fig. 6(c)). 20455 sampled vertices were searched to the closest vertices on ground truth. Minimum of 0.0 (red), maximum of 0.124484 (blue), mean of 0.012780 and RMS 0.023629. Values in the mesh units and concerning the diagonal of the bounding box of the mesh. The main limitation of the results was the bottom part, which was inferred only by the depth sensor.

methodology reduces the incidence of noise and mitigates the low amount of details from depth maps acquired using low-cost RGB-D hardware. Photogrammetry despite capturing a higher level of detail has certain limitations related to the number of resources, like geometric and feature details.

The texturing process using high definition images from SFM output, adding possible missing parts, if needed, also helps to achieve greater visual realism to the reconstructed 3D model.

Future research involves a quantitative analysis of the 3D reconstruction after the texturing step. It is also projected an automation to align point clouds using the scale-based iterative closest point algorithm (scaled PCA-ICP) and the application of this pipeline to digital preservation of artifacts from the cultural heritage of the MAE/UFBA.

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
