# OpenReview forum: "Improved Low-cost 3D Reconstruction Pipeline by Merging Data From Different Color and Depth Cameras"
_graphicsinterface.org/Graphics_Interface/2021/Conference/Second_Cycle — Reject_

### Official Review · Reviewer_1aiE · 2021-05-03
**A paper on improving shape reconstruction by combining data from color and depth cameras. Paper suffers from poor presentation and insufficient evaluation.**

**Rating:** 3
**Confidence:** 5

**Review:**

**Paper Summary**

This paper's main claimed contribution is the system for reconstructing shapes by combining measurements from a depth camera like Microsoft Kinect and pointclouds obtained by using SfM and MVS techniques.

**Strengths**

In terms of paper strengths, the idea of using a low-cost depth camera to obtain coarse depth measurements which are then augmented with the results of SfM/MVS methods is sound.
I do appreciate the effort of creating systems that can fuse the measurements from different sources.
I also do like the fact that the authors have used a 3D printed object to measure the quality of reconstruction. While I have some issues with how the evaluation was performed, I think that the overall paradigm needs to be commended.

**Weaknesses**

Sadly, this paper has a number problems in its current form. My two main reasons for rejecting the paper are:

1. Poor exposition, which makes me question the paper's reproducibility
2. Lack of any comparison to any prior algorithm

Regarding the first point: This paper presents itself as a "step-by-step description of how to overcome the limitations from low-cost three-dimensional data capture". This claim is not well supported by the rest of the text - the paper fails to present each of its steps clearly or to provide sufficient detail. Below I present some concrete examples of the clarity issues:

- section 3.2: "After pairwise incremental registration, an algorithm for global minimization of the accumulated error is run. (...)" Neither the energy function nor the optimization algorithm is mentioned.
- section 3.2: "(...) we use the point cloud obtained by photogrammetry as an auxiliary to apply a new alignment over the depth sensors point clouds, distorting the transformation, propagating the accumulation of errors between consecutive alignments and the loop closure, improving the global registration and the quality of the aligned point cloud." - It is extremely hard to understand what concrete procedures are being performed here. The text claims that pointclouds from photogrammetry are somehow utilized to modify registered depth images from a depth camera, but no concrete detail is provided. The text is missing any equations and/or explanation on how such an optimization problem is set up and solved.
- section 3.2: "(...) it is sufficient just to scale and transform a single capture to fit the cloud obtained by MVS and apply the same transformation to the others (...)" How is this transformation obtained - it is a global registration problem, so what methodology has been used to compute the alignment between reconstructions from two sources? Or is this a purely manual process?
- section 3.4: does not explain the details of the procedure that is being performed. Poses for new photographs are calculated using SfM, but how are they registered to the fused model? Moreover, how is the need for additional photos determined? Is it a purely manual process? The text is also very vague regarding how the texturing itself is performed. The reader needs to refer to section 4 to find out that authors use the approach by Waechter et al.
- Figures 4 and 5 mentions downsampling and filtering steps performed on the output of MVS, but the text is missing any explanation of how these steps are performed.

Regarding the second point: The paper does not provide any comparison to previous work, even though in the related work section the previous work is presented in a way that suggests a need for an improvement. (Section 2 - "The Kinect sensor has considerable limitations, including temporal inconsistency (...). Real-time reconstruction is not a requirement for well-detailed, accurate, and complete reconstruction"). Such wording suggests that the proposed technique produces higher quality results at the cost of runtime performance. It is then necessary to quantify the improvements of the proposed method in comparison to the previous work. However, no effort is being made to compare this work to any publicly available implementations of KinectFusion-like system (for example [VoxelHashing [Niessner' 13](https://github.com/niessner/VoxelHashing)] or the [implementation by Christian Diller](https://github.com/chrdiller/KinectFusionLib)).

Moreover, it is troubling that qualitatively the results do not compare favorably to the KinectFusion system, and without proper comparison on the same set of objects, it is hard to be convinced that a KinectFusion-like system would not be able to produce comparable or better results.

**Summary**

Given the shortcomings I have listed, I cannot recommend this paper for acceptance. I think the paradigm of combining depth measurements with pointclouds obtained from SfM techniques is a good direction, this work, in its current form does not
present enough detail to inform the reader on how exactly the system was built. Moreover, the benefits of the proposed system
over other solutions are not presented.

**Aside**

Additionally, I would like to point out that the Hausdorff distance seems to be simply computed by calling a filter in Meshlab (I make this claim based on the fact that the reported values of minimum, maximum, mean, and RMS are the exact values reported by Matlab's implementation). However, Meshlab computes only a single side of the Hausdorff distance, so the reported values do not follow the mathematical definition of Hausdorff distance. Reference: https://meshlabstuff.blogspot.com/2010/01/measuring-difference-between-two-meshes.html

---

### Official Review · Reviewer_U52R · 2021-05-04
**no sufficient novelty**

**Rating:** 4
**Confidence:** 5

**Review:**

This work describes a 3D reconstruction pipeline using RGBD data. The pipeline is reasonable and leads to plausible results, however, I do not think there is enough novelty for a research paper. The paper solely relies on existing techniques and combines them in a very straightforward manner, so there are no algorithmic or "systems" contributions. The main differentiating factor seems to be that this work advocates to combine high-res imagery with low-res depth. But apart from using SR techniques in one of the steps of the pipeline, it is not clear what are the unique features of the proposed pipeline that would improve its performance with this "hybrid" data.

---

### Official Review · Reviewer_jhtt · 2021-05-04
**This work describes an offline 3D reconstruction pipeline using a cellphone camera for input images and a Kinect depth sensor. The manuscript requires significant revision to improve the quality and clarity of the content.  There also does not appear to be anything sufficiently groundbreaking to support its publication in the current form.**

**Rating:** 4
**Confidence:** 3

**Review:**

The submission attempts to outline a low-cost offline 3D reconstruction pipeline, as the primary contribution, with the use of data from multiple sensors; It makes use of a first-generation Kinect sensor, for depth information, and RGB images from a cellphone camera (Redmi Note 8) along with some run-of-the-mill use of domain tools and libraries. The manuscript is somewhat hard to read at times and the work comes across as rather derivative without any strong contributions. COLMAP and OpenMVS are used for Structure-from-Motion and Multi-View Stereo to achieve a dense reconstruction; The point clouds are aligned, cleaned and fused with some manual intervention. A mesh is generated using Screened Poisson Reconstruction in Meshlab and subsequently textured with image data without going into many details in the manuscript. The approach is tested on a rather small set of six tabletop figurines. A Hausdorff distance comparison, against a ground truth 3d printed source mesh, is used to claim that the combined sensor approach of the pipeline produces improved reconstruction results over the outcome of MVS or SR independently.

Comments:
* The radius from 3D points centroid approach to isolate the target object does not appear to be particularly robust, requiring human intervention.
* The size range of the small number of captured objects is rather limited, given that the captured objects appear to be mostly limited to tabletop figurines, and it remains to be seen how well this will work in general without manual per object tweaking.
* There is lack of a thorough ablation study for the end-to-end system.
* The speed and overall quality of the described offline pipeline is not clearly compared with other prior work in the domain.

Some additional minor corrections/improvements for the manuscript:
* Introduction section: Consider revising the first sentence in the introduction for an improved sentence structure.
* Introduction (5th paragraph): "are disadvantage" -> "are a disadvantage"
* Introduction (7th paragraph): "Such limitations of each data acquisition approach are bypass," -> "Such that limitations ... are bypassed"
* Relevant Work (paragraph 5): "large scales scenes" -> "large scale scenes"
* Section 3.1 (First sentence): "sensor is established" -> "sensor we established"
* Section 3.1 (2nd paragraph): "The depth capture ... to the better the captures by the device":  Consider revising the sentence to improve the sentence structure.
* Section 4 (Texture Synthesis): "The texture synthesis ... comprises the combination" -> "comprises of the combination"
* Figure 5: "This factor makes difficult the reconstruction process by SFM and MVS" -> "This factor makes the reconstruction process by SFM and MVS difficult."
* Experiments and Evaluation (paragraph 6): "when it try to describe featureless regions" -> "when it tries to describe featureless regions"
* Experiments and Evaluation (paragraph 12): "…SFM system did not be able to apply a texture" -> "The SFM system has not been able". Consider revising the entire sentence.
* Figure 7: The two images in the figure could use clearer captioning.

---

### Meta-Review · Area_Chair_oeXb · 2021-05-05

**Recommendation:** Reject
**Confidence:** 5

**Metareview:**

Reviewers uniformly agree that this submission is below the acceptance threshold. While all the reviewers agree that the proposed pipeline is reasonable, however, they also note some significant issues:

1. Lack of algorithmic or "systems" novelty - none of the pipeline steps presented in the paper are novel. The proposed pipeline largely combines existing techniques in a standard manner
2. Clarity issues - paper does not provide enough detail in the description of performed steps, nor does it motivate one choice over another
3. Evaluation issues - paper does not compare with any other work in this domain, nor does it perform a thorough ablation study for different components of the system.

Given that all the reviewers are in agreement, the recommendation is to reject this submission.

---

### Decision · Program_Chairs · 2021-05-08

Reject